# Physician Workforce in Lithuania: Changes during Thirty Years of Independence

**DOI:** 10.3390/healthcare12101023

**Published:** 2024-05-15

**Authors:** Linas Šablinskas, Mindaugas Stankūnas

**Affiliations:** Department of Health Management, Lithuanian University of Health Sciences, 44307 Kauno, Lithuania; mindaugas.stankunas@lsmu.lt

**Keywords:** healthcare system, healthcare planning, healthcare professionals, health resources, Lithuania

## Abstract

Healthcare human resource planning is one of the biggest challenges facing the healthcare systems in many countries. Inadequate decisions in human resource planning can lead to an insufficient number of healthcare professionals then healthcare inequalities. One of the components of resource planning in the healthcare system is long-term data monitoring and the identification of potential trends. Since 1990, the number of physicians in Lithuania has decreased by 15.3% (−2266), but the decrease in the population has led to a 13.61% increase in the number of physicians per 10,000 inhabitants (5.32). During the analyzed period, the largest decrease in the number of physicians workforce by specialty was the number of medical physicians (−73.08%), epidemiology and hygiene (−69.30%), children’s diseases (−49.08%), the most increased number was of family/general practitioners (GPs), geneticists, physical medicine, and rehabilitation specialists. Since 1992, the number of visits to physicians in Lithuania, which has been decreasing for a long time, began increasing, and in 2022 (9.3 visits) it has almost reached the number of visits (9.5) per capita as in 1991. The aim of this research was to collect long-term data from various databases, summarize them, and identify possible trends and the reasons for data changes. The study analyzed data from the Lithuanian healthcare system from the Declaration of Independence of Lithuania to the last 30 years. The data includes or affects the indicators of the healthcare system, changes in population and doctors, the number of visits to doctors, the number of medical students and residents, and data determining inequalities in the healthcare system. Long-term data analysis is useful for developing a model of healthcare human resource planning and for planning healthcare resources.

## 1. Introduction

Lithuania and most other post-communist countries have inherited the Semashko healthcare organization model (named after Nikolai Semashko, a Soviet People’s Commissar for Healthcare), which was characterized by a high level of centralized government administration and the absence of a private healthcare sector. The employees working in the system were civil servants, focusing not on primary healthcare services and the quality of services but on the number of hospitals and physicians [1,2]. This has led to a decline in the average life expectancy of the population and a widening gap between the average life expectancy of women and men [3] compared to other Western countries with a high number of physicians per 10,000 inhabitants and other healthcare challenges [4]. However, on 11 March 1990, after the restoration of Lithuania’s independence, important reforms in various areas of the public sector began [5]. Since the Declaration of Independence, the healthcare system has been undergoing fundamental reforms aimed at improving the health and quality of the services provided by the population, as well as enabling consumer choice. This led to greater attention being paid to primary healthcare and training of family/general practitioners (GPs) in Lithuania [1,6].

The National Concept of Health for Lithuania (1991), approved by the Lithuanian parliament, set priorities for healthcare system reforms and became the basis for the formation of Lithuanian health policy [7]. Ongoing healthcare reform has led to changes in the health insurance system, i.e., when the health insurance system is financed from insurance premiums and collected taxes, the State Health Insurance Fund became responsible for the distribution of healthcare resources, and the Ministry of Health (MoH) is now responsible for the general regulation and organization of the healthcare system. Changes in primary care have enabled not only the development of the public sector but also the private sector, thus extending the freedom of choice to consumers and the extremely rapid growth of the number of GPs [6].

It was already clear in 1991 that the number of physicians in Lithuania was significantly higher than in other European countries [8]. This is what decided the need to determine the optimal number of physicians in Lithuania. In 1994, the first working group was set up, which presented an assessment that there are too many physicians in Lithuania, the distribution of physicians by specialty is uneven, and the shortage of physicians in the regional areas is particularly high [9]. However, the subsequent events (especially the increase in the migration of physicians after accession to the European Union (EU)) [10] led to the opposite trend, and the number of newly admitted students in medical studies has increased [3,11,12]. The result of this increase is that Lithuania currently has one of the largest numbers of physicians among the EU countries, but experts tend to believe that the shortage of physicians in the future will become an urgent problem [13]. In addition to the growing general demand for healthcare professionals, the geographical and specialty/profession imbalance in Lithuania is also an issue of concern [14]. According to the information of the Government Strategic Analysis Centre, in 2020, the greatest shortage of physicians was GPs, internal disease, infectious diseases (this choice could be determined by COVID-19), cardiologists, anesthesiologists—reanimatologists. In the future (2030), the projected greatest shortage will be of GPs and internal diseases [15]. This problem is particularly acute in rural areas, leading to lower healthcare access [1]. Other problems include a lack of healthcare system funding and a rather high focus of healthcare services on specialized and in-patient services contribute to the Lithuanian healthcare system receives poor evaluations among other Central and Eastern European countries [16].

When analyzing the human resources situation in Lithuanian healthcare, it is important to evaluate not only the changes in population and physicians, the number of visits to physicians, and the number of medical students and residents but also the data that determine the inequalities. These assessments can contribute to the development of the human resource planning model for healthcare, as long-term data analysis can help assess future trends and develop reliable forecasting models.

This paper provides an overview of trends and reforms in the physician workforce since Lithuania gained independence in 1991. This paper’s main audience comprises those interested in the health systems of the former Soviet countries. However, this analysis offers lessons for other countries in Europe and elsewhere as it presents transitions in the healthcare workforce during the rapid transformation of the health system and society in general.

## 2. Materials and Methods

For the analysis of trends in the physicians’ workforce in Lithuania in the post-independence period (since 1991), we used different sources of secondary (aggregated) data:World Health Organization (WHO) database for obtaining data on the number of physicians per 10,000 inhabitants of Lithuania [8].Institute of Hygiene information sources (published reports for 1991–1998; internal electronic database for 1999–2021) for obtaining information about the number of medical and residency students who graduated medical studies in Lithuanian universities (Lithuanian University of Health Sciences, and Vilnius University), the total number of all physicians and physicians by specialty and county, number of visits to physicians, and number of visits to physicians per one inhabitant [3,11].State Data Agency of Lithuania database for obtaining data on the number of population [17].

Data on changes in the number of physicians are presented following the order of the Minister of Health of the Republic of Lithuania of 1999, in which figures in terms of higher education in medicine (physicians’ qualification) are divided into medical doctor’s practice, 19 specialties, and 42 sub-specialties, as detailed in the table below (Table 1) [18].

List of personal healthcare specialties and sub-specialties.

The data were entered into a Statistical Package for the Social Sciences (SPSS) program version 20 database. The variables included: the total number of physicians, number of physicians per 10,000 inhabitants, number of physicians by specialty and county, number of visits to physicians, number of visits to physicians per one inhabitant, and number of graduates in medical studies.

A simple descriptive statistical analysis was performed using the SPSS 22.0 and MS Office Excel 2019 software package. Trends in physician supply and number of visits/consultations were analyzed using the method of linear regression. Coefficients of regression (b) multiplied by 100 were presented as average annual changes (AAC), with 95% confidence intervals (CI). A *p*-value < 0.05 was considered statistically significant. Ethical approval was not required, as only secondary (aggregated) data were used for this study.

## 3. Results

The total number of physicians in Lithuania has decreased by 15.3% (−2266) since 1990. In the same period, the population of Lithuania decreased (−24.4%, or by 903,700 inhabitants) [3,11]. This has led to the fact that the number of physicians per 10,000 inhabitants increased by 13.61% (5.32) during this period. However, over the 30 years, this figure ranged from 39.1 (in 1990) to the lowest rate of 34.93 (in 1992), increasing to 36.37 (in 1997) and again to 35.78 (in 2004). From 2005, it began to increase from 36.64 to 44.81 (in 2019). According to the latest data from the WHO, this number has increased to 49.5 (2021), also reaching the maximum number of physicians per 10,000 inhabitants over the entire data reporting period since 1990 (Figure 1) [8].

Since 1991, the number of physicians has changed mostly in all specialties. The most significant decrease was observed among these groups of physicians: epidemiology and hygiene (AAC = −3.98 [95% CI, −4.73; −3.23]), pediatrics (AAC = −2.59 [95% CI, −2.84; −2.33]), internal medicine (AAC = −2.33 [95% CI, −2.58; −2.07]) and laboratory medicine (AAC = −2.07 [95% CI, −2.70; −1.45]), obstetrics and gynecology (AAC = −1.38 [95% CI, −1.50; −1.27]). In other specialties such as otorhinolaryngology (AAC = −0.83 [95% CI, −0.95; −0.70]), ophthalmology (AAC = −0.26 [95% CI, −0.39; −0.12]), forensic medicine (AAC = −0.89 [95% CI, −1.37; −0.41]), and surgery (AAC = −0.07 [95% CI, −0.22; 0.08]), reductions were recorded, but not as high (Table 2).

The number of GPs increased the most from 81 specialists in 1993 to 2043 specialists in 2021 (AAC = 10.77 [95% CI, 8.00; 13.54]). Following resident physicians (AAC = 2.16 [95% CI, 1.55; 2.77]), geneticists (AAC = 6.02 [95% CI, 4.37; 7.68]), physical medicine and rehabilitation (AAC = 2.03 [95% CI, 1.35; 2.70]), healthcare administrators and statisticians (AAC = 3.48 [95% CI, 2.10; 4.85]), anesthesiologists (AAC = 1.36 [95% CI, 1.13; 1.58), psychiatrists (AAC = 0.53 [95% CI, 0.27; 0.78]), orthopedists traumatologists (AAC = 1.24 [95% CI, 0.98; 1.49]) and radiologists (AAC = 0.18 [95% CI, 0.03; 0.33]) (Table 2).

There are significant differences in the number of physicians between cities and rural areas in Lithuania. This difference has been increasing since 2003. The number of specialists working in cities remains similar, but in rural areas, this number tends to be decreasing (Figure 2).

The largest number of physicians in the 1995–2022 period was recorded in the counties of Vilnius and Kaunas (Appendix A Table A1: The number of active physicians working in the MoH system and in the private sector in 1995–2022 per 10,000 inhabitants, by county [3,11]).

The absolute number of physicians’ visits in Lithuania during the 1991–2018 period decreased from 35.37 million to 25.44 million (AAC = −0.84 [95% CI, –1.52; −0.17]). However, a negative population growth led to the number of visits per 1 inhabitant almost reaching the numbers of 1991 (Table 3).

In 2019, Lithuania introduced a remote consultation of physicians. These consultations can be provided for acute illnesses such as: colds, fever, urinary tract infections, etc., as well as non-urgent medical problems that do not require a physical examination [19]. Remote consultations in 2019 accounted for 0.69% of all visits, 2020—27.16%, 2021—31.23%, and 2022—21.24% (Table 3). The growth of remote consultations and the decrease in the number of contact visits was influenced by the COVID pandemic that began in 2020 and the restrictions imposed by the Government of the Republic of Lithuania, which led to the restriction of contact visits to healthcare professionals or making them entirely impossible [20].

**Table 3 healthcare-12-01023-t003:** The number of visits to physicians in Lithuania 1991–2022 (1991–2000 [21] and 2003–2022 [22]) every three years.

Year	Number of Visits to Physicians (Million)	The Number of Visits to Physicians Is per One Inhabitant
1991	35.37	9.5
1994	28.90	7.8
1997	26.74	7.2
2000	22.46	6.4
2003	19.58	5.7
2006	20.94	6.4
2009	21.81	6.9
2012	22.52	7.5
2015	24.22	8.3
2018	25.44	9.1
1991–2018 (AAC [95% CI])	−0.84 [−1.52; −0.17] *	0.31 [−0.42; 1.03]
2021	24.79 (7.74 ^†^)	8.8 (2.8 ^†^)
2022	26.40 (5.59 ^†^)	9.3 (2.0 ^†^)

^†^—Percentage of remote consultations; *—*p* < 0.05.

The largest increase in visits at the first level was to GPs from 3.05 million (in 2001) to 12.4 million (2018) (AAC = 6.24 [95% CI, 4.47; 8.01]). The largest decrease was in visits to district therapists and surgeons (AAC = −6.83 [95% CI, −7.60; −6.07] and AAC = −5.28 [95% CI, −5.67; −4.90] accordingly). The total number of visits at the first level increased by 36.29% (AAC = 1.50 [95%CI, 1.15; 1.85]) (Appendix A Table A2: The number of visits to physicians in Lithuania in 2001–2022 according to the long list of specialties [22]).

The number of visits to physicians in the second and third levels also increased by 37.33% (AAC = 2.05 [95% CI, 1.68; 2.42]) (2001–2018). The highest increase was to anesthesiologists-reanimatologists (AAC = 17.11 [95% CI, 15.57; 18.65]), geneticists (AAC = 10.97 [95% CI, 9.71; 12.24]), psychotherapists (AAC = 5.47 [95% CI, −0.34; 11.28]), plastic and reconstructive surgeons (AAC = 8.82 [95% CI, 7.23; 10.42]) and vascular surgeons (AAC = 8.48 [95% CI, 7.57; 9.39]). The greatest decrease in visits was to physicians such as phthisiatricians (AAC = −31.71 [95% CI, −34.62; −28.81]), oncologists (AAC = −34.28 [95% CI, −43.11; −25.45]), cardiac surgeons (AAC = −2.71 [95% CI, −4.98; −0.43]), chest surgeons (AAC = −0.40 [95% CI, −2.12; 1.33]), otorhinolaryngologists (AAC = −2.26 [95% CI, −2.68; −1.84]), these reductions could also be due to the emergence of new specialists who took over some of the functions and at the same time visits (Appendix A Table A2).

Although remote consultations appeared in 2019 and were only at the first level, accounting for only 1.06% of all first-level appointments, the number of remote consultations increased significantly due to the COVID-19 pandemic, reaching 27.24% in 2020, 31.23% in 2021, 21.16% in 2022 [22].

Between 1991 and 2021, 11,576 medical students graduated from medical studies, and 12,036 individuals completed residency studies [3,11]. However, the annual number of graduates has differed over this period. In 1991, there were 284 medical graduates and in 1992—373. Since 1994, medical graduations started to decrease reaching the lowest value of 216 in 2005 (Figure 3). In 2002 the decision was made to increase the admission to medical training to 400 students annually [12]. This important decision was made based on previous planning forecasts [23]. As a result, the number of students graduating from medical studies has increased. The highest number of medical graduates was recorded in 2020 and 2021 when the number of graduates reached 571 in both years and compared to 1991, this number increased by 201.01%.

The first data on the number of individuals who completed medical residency is from 1996, when the number of individuals who completed residency studies was 516. In 1997, the highest number (704) of successful medical residency graduates was recorded since the Declaration of Independence. The lowest number was recorded in 2011—233, respectively, due to the low number of medical graduates in 2005. Since 2013, the number of individuals who graduated from medical residency has increased, so it has fluctuated around 400 (Figure 3).

## 4. Discussion

Although the global health workforce is growing, this does not guarantee a sufficient number of healthcare professionals. According to The Global Strategy on Human Resources for Health: Workforce 2030 adopted by the WHO, the shortage of healthcare professionals was 17.4 million in 2013 and will remain at 14.5 million in 2030. The shortage of specialists in Europe is projected to be like that in 2013 and will reach 0.1 million in 2030. In 2022, it was proposed to recalculate this figure, considering that in 2020 there were around 65 million healthcare professionals worldwide (data from 194 member states), which represents an increase of 29% since the adoption of the strategy in 2016; however, the shortage of healthcare professionals will remain at 15 million in 2022, and at 10 million in 2030 [24]. Similar challenges are common for the Baltic States.

Estonia, Latvia, and Lithuania were consistently among the leaders in the EU countries, according to number of physicians per 10,000 inhabitants, but even 30 years after the restoration of independence, the number of physicians decreased only in Latvia by 1.65 physicians, in Estonia the number increased by 3.63 physicians, in Lithuania by 5.34 physicians for 10,000 inhabitants. (Table 4), this is also due to the declining population in all three countries [8]

The present number of physicians in Lithuania is high, but shortages are predicted. The most recent attempt to project the supply of physicians in Lithuania was done by STRATA. This projection suggested that Lithuania will be in an extreme shortage of the following physicians by 2030: general practitioners (−428), internal medicine (−420), pediatrics (−317), emergency medicine (–256), and anesthesiologists (–143) [15].

The COVID-19 pandemic, which began in 2020, had a significant impact on demographic and healthcare indicators. Although the increase in life expectancy between 2010 and 2019 in Lithuania was the fastest in the EU, the impact of COVID-19 was a major setback. Life expectancy in Lithuania in 2020 was the third lowest in the EU and 5.5 years below the EU average [25]. The number of visits to physicians has decreased significantly, not due to the improvement of the health of the population but to restrictions imposed by the Government of the Republic of Lithuania [20]. According to the number of visits at the primary level of healthcare, the number of visits not only reached the pandemic level but also significantly exceeded it. On the second and third levels, the number of visits is also growing rapidly (Appendix A Table A1). Such rapid growth will also impact the planning of human healthcare resources.

The results from our study and information from other countries suggest the need for continuous and long-term planning of health human resources. However, the planning of the health workforce requires a deep understanding of processes and dynamics in the field. If these processes are ignored, misleading interpretations and decisions can be made. For instance, the difference between physicians in urban and rural areas has significantly decreased in Lithuania in 2011. Nevertheless, this shift was not due to any positive changes in the geographical imbalance of physicians but rather to changes in the MoH regulation and status of resident physicians [3,11]. Another example is that even though significant reductions were recorded among medical practitioners and internal medicine professionals in Lithuania, this does not mean that these professionals left the labor market. This was due to the healthcare reforms, which provided an opportunity for these specialists to re-train and become general practitioners (GPs) [26].

Another challenge for health workforce planning in Lithuania is the absence of unified database, which stores all information about health human resources in the country. This means that data must be collected from a variety of databases or reports, where calculation methodologies or data sources do not always coincide. Moreover, some databases have limited public access. These are some of the reasons why healthcare human resource planning is quite complicated, as it limits the observation of past trends and complicates the development of models for the planning of health human resources.

We must address some possible limitations of this paper. The data analyzed covered a long period (30 years), which may have led to changes in the methodology of data collection and affected the accuracy of the data. Also, the analyzed databases contain data collected from healthcare organizations, some of which may have provided inaccurate data or no data at all.

## 5. Conclusions

Our data analysis suggests that the number of specialist physicians in Lithuania has been changing due to policy decisions and the increasing number of visits to physicians. Although the number of physicians in Lithuania is high, the above-mentioned reasons may lead to a shortage in the future, especially in the rural areas. Achieving a balance between the supply and demand of physicians is a very complex but important task to ensure the effective performance of the health system and accessibility to health services. This healthcare challenge is expected not only in Lithuania and the Baltic States but also in other countries. Therefore, further research in this area and the exchange of the best practices between countries is critically important.

## Figures and Tables

**Figure 1 healthcare-12-01023-f001:**
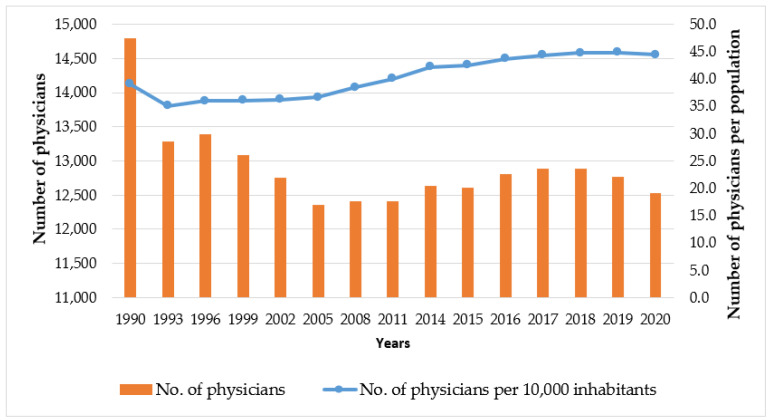
Physicians’ number/per population (over 10,000) ratios in Lithuania every three years in the 1990–2020 period [8].

**Figure 2 healthcare-12-01023-f002:**
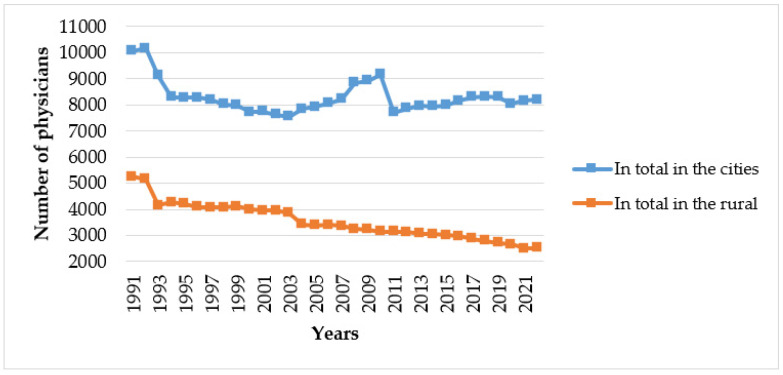
Number of active physicians working in cities and rural areas, 1991–2022 [2,7].

**Figure 3 healthcare-12-01023-f003:**
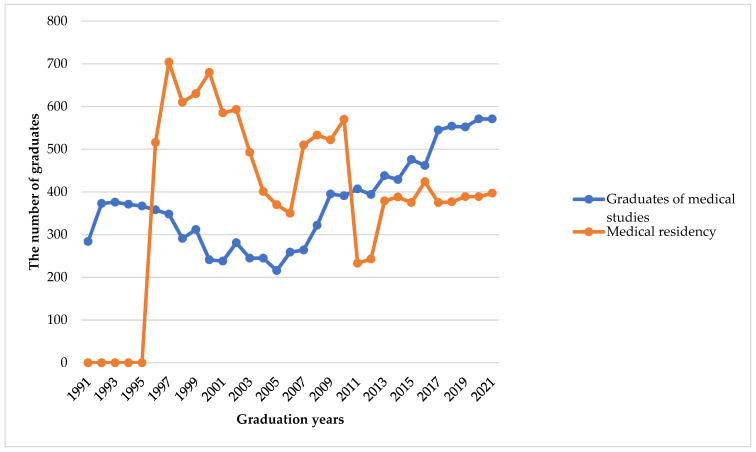
The number of students who completed medical studies and medical residency studies at the Lithuanian University of Health Sciences and Vilnius University during the period of 1991–2021 [3,11].

**Table 1 healthcare-12-01023-t001:** Nineteen specialties with sub-specialties are grouped together from the list of 61 specialties approved by the Ministry of Health [18].

Medical Doctor’s Practice	
Specialties	Sub-Specialties
General practitioners (GPs)	
Internal medicine	Allergology and clinical immunology; occupational medicine; dermatovenerology; endocrinology; geriatrics; hematology; infectology; cardiology; nephrology; neurology; pulmonology; rheumatology; clinical toxicology; oncology chemotherapy; oncology radiotherapy; intensive care medicine.
Pediatrics	Neonatology; pediatric endocrinology; pediatric gastroenterology; pediatric hematology; pediatric cardiology; pediatric nephrology; pediatric neurology; pediatric pulmonology; pediatric intensive care medicine.
Dietetics	
Surgery	Abdominal surgery; plastic and reconstructive surgery; vascular surgery; heart surgery; neurosurgery; chest surgery; urology.
Orthopedics traumatology	
Pediatric surgery	
Anesthesiology reanimation	
Obstetrics gynecology	
Ophthalmology	
Otorhinolaryngology	
Psychiatry	Child and adolescent psychiatry.
Genetics	
Forensic medicine	
Physical medicine and rehabilitation	Sports medicine.
Pathology	
Laboratory medicine	
Radiology	
General dental practice	Oral surgery; orthodontic dentistry, endodontology, orthopedic dentistry; periodontology; pediatric dentistry; maxillofacial surgery.

**Table 2 healthcare-12-01023-t002:** Number of physicians in Lithuania by specialties: 1991–2021 every five years [3,11].

The Number of Active Physicians According to Specialties	(Per 10,000 Persons and (Natural Persons))							AAC [95% CI] **
	1991	1996	2001	2006	2011	2016	2021	
General practitioners	n.d.	0.50 (181)	2.58 (897)	5.30 (1794)	6.19 (1860)	7.37 (2099)	7.35 (2043)	11.95 [9.22; 14.67] * (10.77 [8.00; 13.54]) *
Internal medicine	12.71 (4757)	12.98 (4656)	10.77 (3744)	9.31 (3151)	9.75 (2930)	9.88 (2813)	9.41 (2561)	−1.16 [−1.46; −0.86] * (−2.33 [−2.58; −2.07]) *
Pediatrics	4.49 (1681)	5.22 (1873)	19.76 ^†^ (1636)	18.26 ^†^ (1270)	20.42 ^†^ (1136)	19.75 ^†^ (1008)	17.64 ^†^ (856)	5.08 [3.35; 6.81] * (−2.59 [−2.84; −2.33]) *
Dietetics	0.06 (21)	0.06 (20)	0.07 (25)	0.06 (19)	0.07 (21)	0.08 (22)	0.08 (23)	1.33 [0.74; 1.91] * (0.11 [−0.48; 0.71])
Surgery	2.45 (915)	2.58 (927)	2.38 (826)	2.55 (864)	2.76 (828)	3.03 (862)	3.25 (870)	1.05 [0.84; 1.26] * (−0.07 [−0.22; 0.08])
Orthopedics traumatology	0.82 (307)	0.88 (317)	0.88 (305)	0.97 (330)	1.12 (336)	1.44 (409)	1.35 (377)	2.44 [2.14; 2.73] * (1.24 [0.98; 1.49]) *
Pediatric surgery	n.d.	0.18 (63)	0.80 ^†^ (66)	0.91 ^†^ (63)	1.31 ^†^ (73)	1.23 ^†^ (63)	1.36 ^†^ (66)	6.69 [4.40; 8.99] * (0.72 [−0.04; 1.48])
Anesthesiology	1.48 (552)	1.61 (576)	1.62 (562)	1.77 (600)	2.29 (687)	2.71 (771)	2.97 (823)	2.54 [2.27; 2.81] * (1.36 [1.13; 1.58]) *
Obstetrics gynecology	2.34 (877)	2.36 (846)	2.33 (810)	2.37 (801)	2.33 (699)	2.27 (647)	2.13 (587)	−0.25 [−0.35; −0.16] * (−1.38 [−1.50; −1.27]) *
Ophthalmology	1.00 (376)	1.01 (362)	1.02 (355)	1.00 (340)	1.16 (348)	1.28 (364)	1.16 (319)	0.90 [0.71; 1.09] * (−0.26 [−0.39; −0.12]) *
Otorhinolaryngology	0.86 (323)	0.88 (315)	0.86 (300)	0.86 (291)	0.92 (275)	0.96 (274)	0.87 (239)	0.33 [0.16; 0.50] * (−0.83 [−0.95; −0.70]) *
Psychiatry	1.21 (454)	1.41 (505)	1.59 (553)	1.75 (594)	1.84 (553)	1.95 (555)	2.03 (562)	1.72 [1.52; 1.92] * (0.53 [0.27; 0.78]) *
Genetics	n.d.	0.02 (8)	0.02 (8)	0.03 (10)	0.04 (13)	0.06 (17)	0.07 (19)	7.70 [5.59; 9.81] * (6.02 [4.37; 7.68]) *
Forensic medicine	0.16 (60)	0.18 (65)	0.22 (78)	0.17 (57)	0.17 (50)	0.18 (51)	0.19 (54)	0.28 [−0.18; 0.74] (−0.89 [−1.37; −0.41]) *
Physical medicine and rehabilitation	0.56 (211)	0.76 (272)	1.13 (394)	1.19 (403)	1.32 (396)	1.46 (415)	1.37 (375)	3.17 [2.53; 3.80] * (2.03 [1.35; 2.70]) *
Pathology	0.18 (68)	0.19 (69)	0.21 (74)	0.19 (66)	0.21 (64)	0.26 (73)	0.25 (70)	0.68 [0.31; 1.05] * (−0.51 [−0.84; −0.18]) *
Laboratory medicine	n.d.	0.43 (153)	0.34 (118)	0.26 (88)	0.31 (93)	0.30 (86)	0.32 (87)	−0.88 [−1.60; −0.17] * (−2.07 [−2.70; −1.45]) *
Radiology	1.23 (459)	1.37 (493)	1.30 (451)	1.42 (481)	1.54 (464)	1.72 (489)	1.79 (492)	1.35 [1.16; 1.54] * (0.18 [0.03; 0.33]) *
Medical doctor’s practice	n.d.	n.d.	0.48 (167)	0.47 (160)	0.47 (141)	1.05 (300)	1.45 (406)	7.96 [6.11; 9.81] * (6.68 [4.80; 8.55]) *
Residency	n.d.	n.d.	3.36 (1168)	3.23 (1092)	5.50 (1653)	5.94 (1692)	6.34 (1773)	3.44 [2.73; 4.14] * (2.16 [1.55; 2.77]) *
Epidemiology and hygiene	0.95 (355)	0.99 (356)	0.77 (267)	0.89 (301)	0.74 (222)	0.66 (189)	0.39 (109)	−2.84 [−3.58; −2.11] * (−3.98 [−4.73; −3.23]) *
Healthcare administrators and statisticians	n.d.	n.d.	0.53 (185)	0.90 (306)	1.28 (383)	1.26 (359)	0.96 (268)	4.76 [3.34; 6.18] * (3.48 [2.10; 4.85]) *
Other	3.76 (1407)	3.34 (1198)	3.00 (1042)	1.27 (429)	0.34 (102)	0.43 (123)	n.d.	−12.28 [−13.91; −10.66] * (−13.46 [−15.13; −11.79]) *

^†^ The calculation methodology has changed, and the number is not provided for all inhabitants, but per 10,000 children up to the age of 14; * *p* < 0.05; ** for 1991 (or earliest available)–2021; n.d.—no data.

**Table 4 healthcare-12-01023-t004:** Physicians’ number/to population (per 10,000) ratios in Estonia, Latvia, and Lithuania every three years in the 1990–2020 period [8].

	Number of Physicians (Persons)			Number of Physicians (per 10,000 Inhabitants)		
	Estonia	Latvia	Lithuania	Estonia	Latvia	Lithuania
1990	5498	9439	14,795	35.0	35.1	39.1
1993	4792	7446	13,285	31.85	28.22	35.07
1996	4457	6966	13,389	31.07	27.49	35.97
1999	4200	6495	13,092	29.93	26.77	36.03
2002	4089	6443	12,749	29.65	27.7	36.26
2005	4257	6628	12,361	31.42	29.68	36.64
2008	4469	7040	12,413	33.43	32.74	38.5
2011	4372	6456	12,407	32.94	31.08	40.03
2014	4418	6412	12,631	33.61	31.89	42.18
2017	4569	6225	12,887	34.68	31.84	44.37
2020	5136	6346	12,529	38.63	33.45	44.42

## Data Availability

Data are contained within the article.

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
