# Peer review of "Physician Workforce in Lithuania: Changes during Thirty Years of Independence"

_healthcare, 2024, doi:10.3390/healthcare12101023_

Round 1

Reviewer 1 Report

Comments and Suggestions for Authors

 Healthcare Professionals in Lithuania: Changes during Thirty Years of Independence

I have reviewed the above mentioned manuscript. The authors attempt to address a pertinent healthcare provision problem of doctor-patient ratios. They also assess the available medical practitioner skills or specialty to inform resource planning in the healthcare system by identifying potential trends. However, I have a few comments for this work as follows:

Major comments:

·         The authors have limited their research problem to only Lithuania and they have failed to highlight the significance of their work to first the European context and the rest of the world in general.

·         If one outside of Lithuania reads this study, how will it benefit them?

·         Research is meant to add new information for the entire scientific community who may be interested in the topic of interest.

·         The methodology is vague and lacks details and the data used had not been adequately described, for instance, what were the data variables that were used in the analysis?

·         The authors allude to forecasting doctor availability even until 2030 yet there was no forecasting model that was described in their methodology, actually forecasting is only mentioned in the discussion.

·         Glaringly so, there was no methodology described in this work, except for only mentioning the data used and their sources. Where and what is the methodology?

·         Merely stating that Excel was used to analyze the data and produce the graphs is not enough as a methodology.

·         The methodology is meant to provide a step by step process of how the analysis work was conducted before results were produced.

·         The presentation of percentage decline or increase cannot constitute statistical analysis, especially when we cannot tell if the decline or increase was significant or not using exact p-values and 95% confidence intervals.

Minor comments

The authors throw statements and terms around without adequately describing them for someone outside Lithuania. For example this was found in:

·         Line 32 of the introduction, what is this healthcare model the authors alluded to?

·         The statement on line 47 of the introduction, how has this National Concept been a major driver of the healthcare system?

·         What are regional areas in line 61 and what are districts in line 74? How are the 2 different?

·         What does special distribution of doctors mean in line 68 to 69?

·         The authors should refrain from making vague and incomplete statements throughout the manuscript.

·         It would help to mention the total number of doctors before presenting the percentages in the results section.

·         The data used need to be described in details including the size of the variables before presentation of the results.

·         Check grammar in line 152 to line 153

·         Explain what is meant by remote visits to doctors in line 160

·         Line 249, this is absurd, what is the strata analysis and why introduce it here?

·         Line 250 and 265, your analysis did not include any forecasting model for the coming years. So where is this estimate coming from?

·         Which region in line 319?

·         Change tense in line 57

·         The capital letters starting each word in the title are not necessary.

Comments on the Quality of English Language

Minor grammatical errors and tense corrections need to be addressed.

Author Response

Dear Reviewer,

On behalf of all authors, I wish to thank you for your time in reading our manuscript and your valuable comments. We hope that these comments and suggestions have been addressed properly. The following corrections and modifications have been completed:

Comment: <The authors have limited their research problem to only Lithuania and they have failed to highlight the significance of their work to first the European context and the rest of the world in general. If one outside of Lithuania reads this study, how will it benefit them? Research is meant to add new information for the entire scientific community who may be interested in the topic of interest.>

Answer: We cannot agree, that this information will be interesting only to Lithuanian audience. This paper could be interesting to researcher and policy makers, who work in the field of health human resource management. We added a short paragraph to “Introduction”, which highlights possible value of this paper to wider international community.

Comment: <The methodology is vague and lacks details and the data used had not been adequately described, for instance, what were the data variables that were used in the analysis?>

Answer: We have extended and improved the description on of methodology.

Comment: <The authors allude to forecasting doctor availability even until 2030 yet there was no forecasting model that was described in their methodology, actually forecasting is only mentioned in the discussion.>

Answer: Thank you for this comment. We agree that some paragraphs and our comments in “Discussion” section could be misleading. We do not make any forecasting of human resources in this manuscript. We made corrections in “Discussion”.

Comment: <Glaringly so, there was no methodology described in this work, except for only mentioning the data used and their sources. Where and what is the methodology? Merely stating that Excel was used to analyze the data and produce the graphs is not enough as a methodology. The methodology is meant to provide a step by step process of how the analysis work was conducted before results were produced. The presentation of percentage decline or increase cannot constitute statistical analysis, especially when we cannot tell if the decline or increase was significant or not using exact p-values and 95% confidence intervals.>

Answer: We made major modifications to the description of the methodology. Moreover, we did more advanced statistical analysis. The average annual changes (with 95 confidence intervals and p-values) were calculated.

Comment: < Line 32 of the introduction, what is this healthcare model the authors alluded to?>

Answer: This model was used for governance of healthcare system in the Soviet Union. We have improved the description of this model in the manuscript

Comment: <The statement on line 47 of the introduction, how has this National Concept been a major driver of the healthcare system?>

Answer: The National Concept of Health is the main health policy document in Lithuania. We have improved the description of this concept in the manuscript.

Comment: <What are regional areas in line 61 and what are districts in line 74? How are the 2 different?>

Answer: We corrected this expression from “regional” and “districts” to “rural areas”

Comment: <What does special distribution of doctors mean in line 68 to 69?>

Answer: We corrected this expression from “special” to “specialty/profession imbalance”.

Comment: <The authors should refrain from making vague and incomplete statements throughout the manuscript.>

Answer: Thank you, we corrected information in the text.

Comment: <It would help to mention the total number of doctors before presenting the percentages in the results section.>

Answer: This information was added in the text (Line 110). The trend of total number of doctors is presented in Figure 1.

Comment: <The data used need to be described in details including the size of the variables before presentation of the results.>

Answer: We corrected "Materials and methods” part.

Comment: <Check grammar in line 152 to line 153>

Answer: We corrected grammar

Comment: <Explain what is meant by remote visits to doctors in line 160>

Answer: We corrected this expression from “remote visits” to “remote consultations” and added a short description: “In 2019, Lithuania has introduced a remote consultation of doctors. These consultations can be provided for acute illnesses such as: cold, fever, urinary tract infection, etc., as well as non-urgent medical problems that do not require a physical examination.”

Comment: <Line 249, this is absurd, what is the strata analysis and why introduce it here?

Answer: We corrected this information

Comment: <Line 250 and 265, your analysis did not include any forecasting model for the coming years. So where is this estimate coming from?>

Answer: This information was added in the text (Line 312-317).

Comment: <Which region in line 319?>

Answer: We corrected this part to “The Baltic States”

Comment: <Change tense in line 57>

Answer: Thank you. We corrected tense (Line 64).

Comment: <The capital letters starting each word in the title are not necessary.>

Answer: Thank you. We corrected the title following your recommendation.

We hope that you will find our improvement appropriate. Once again, thank you for all your recommendations. They were very helpful.

Reviewer 2 Report

Comments and Suggestions for Authors

1.      I acknowledge the motivation behind this study and recognize its contribution as a reference point for analyzing the healthcare landscape in Lithuania. The data sourced from the WHO and library are considered robust. However, a critical concern arises regarding the measures taken to ensure the reliability of the methodology retrieving these data. What steps were taken to verify that the retrieved data accurately reflect the state of medical healthcare in Lithuania for both 1990 and 2022? Are there additional resources available to cross-reference the data presented here? Section 2 requires improvement; readers may seek a detailed methodology outlining how the experiment was established and how the data retrieval process was conducted.

2.      It is essential to address the limitations of this study by including a dedicated section to highlight what aspects were not covered. This "Limitations" section will provide clarity on the scope of the study and help readers understand its constraints.

3.      The conclusion section warrants improvement as it currently lacks depth and fails to fully reflect the content covered in the manuscript. Furthermore, the final sentence appears incomplete, leaving readers with unanswered questions. Enhancements to this section are necessary to ensure that it effectively summarizes the key findings and insights presented throughout the manuscript.

Overall, while the data retrieved from reputable organizations are commendable, it is imperative for the authors to elucidate the methodologies employed to ensure the reliability of the data. Additionally, detailing the limitations of the study will aid policymakers in making informed decisions when utilizing this manuscript to support decision-making processes in other contexts.

Author Response

Dear Reviewer,

On behalf of all authors, I wish to thank you for your time in reading our manuscript and your valuable comments. We hope that these comments and suggestions have been addressed properly. The following corrections and modifications have been completed:

Comment: <I acknowledge the motivation behind this study and recognize its contribution as a reference point for analyzing the healthcare landscape in Lithuania. The data sourced from the WHO and library are considered robust. However, a critical concern arises regarding the measures taken to ensure the reliability of the methodology retrieving these data. What steps were taken to verify that the retrieved data accurately reflect the state of medical healthcare in Lithuania for both 1990 and 2022? Are there additional resources available to cross-reference the data presented here? Section 2 requires improvement; readers may seek a detailed methodology outlining how the experiment was established and how the data retrieval process was conducted.>

Answer: We made major modifications to the description of the methodology. Moreover, we did more advanced statistical analysis. The average annual changes (with 95 confidence intervals and p-values) were calculated. Therefore “Results” section looks now quite differently.

Comment: <It is essential to address the limitations of this study by including a dedicated section to highlight what aspects were not covered. This "Limitations" section will provide clarity on the scope of the study and help readers understand its constraints. This study has some methodological issues requiring explanation.>

Answer: We added a short paragraph about main methodological limitations (line 386-390).

Comment: <The conclusion section warrants improvement as it currently lacks depth and fails to fully reflect the content covered in the manuscript. Furthermore, the final sentence appears incomplete, leaving readers with unanswered questions. Enhancements to this section are necessary to ensure that it effectively summarizes the key findings and insights presented throughout the manuscript.>

Answer: We corrected "Conclusion” part.

Comment: <Overall, while the data retrieved from reputable organizations are commendable, it is imperative for the authors to elucidate the methodologies employed to ensure the reliability of the data. Additionally, detailing the limitations of the study will aid policymakers in making informed decisions when utilizing this manuscript to support decision-making processes in other contexts.>

Answer: We corrected "Materials and methods” part and added information about limitations in the text Line 386-390.

We hope that you will find our improvement appropriate. Once again, thank you for all your recommendations. They were very helpful.

Reviewer 3 Report

Comments and Suggestions for Authors

Dear authors,

thank you for the opportunity to review your manuscript. I would recommend to enlarge the introduction and theoretical background section. The materials and methods are described quite well but your results and conclusions are not very contributive or novel. They are only describing the clear phenomena without providing additional insights on this topic or solutions to solve the issues of the Lithuanian healthcare system. With the wide range of datasets I would expect the prediction of relationships and their verification by the use of statistical methods.

Wish you all the best 

Author Response

Dear Reviewer,

On behalf of all authors, I wish to thank you for your time in reading our manuscript and your valuable comments. We hope that these comments and suggestions have been addressed properly. The following corrections and modifications have been completed:

Comment: <thank you for the opportunity to review your manuscript. I would recommend to enlarge the introduction and theoretical background section. The materials and methods are described quite well but your results and conclusions are not very contributive or novel. They are only describing the clear phenomena without providing additional insights on this topic or solutions to solve the issues of the Lithuanian healthcare system. With the wide range of datasets I would expect the prediction of relationships and their verification by the use of statistical methods.>

Answer: We corrected "Materials and methods” part and added information in “Results” part. In addition, we have made some modification in “Introduction”

We hope that you will find our improvement appropriate. Once again, thank you for all your recommendations. They were very helpful.

Round 2

Reviewer 1 Report

Comments and Suggestions for Authors

The authors have addressed all my comments that were raised in the original version of the manuscript. I am happy with changes that were made in the paper.

Comments on the Quality of English Language

None

Reviewer 3 Report

Comments and Suggestions for Authors

Dear Authors,

thank you for the opportunity to review you manuscript. I recommend to add the interpretation (not only presentation) of your statistic results (AAC and CI) into the results section to make your manuscript more attractive for readers. The future research orientation would be also contributive in the end of the paper.

Wish you good luck